# The Metabolic Efficacy of a Cannabidiolic Acid (CBDA) Derivative in Treating Diet- and Genetic-Induced Obesity

**DOI:** 10.3390/ijms23105610

**Published:** 2022-05-17

**Authors:** Elad Ben-Cnaan, Anna Permyakova, Shahar Azar, Shira Hirsch, Saja Baraghithy, Liad Hinden, Joseph Tam

**Affiliations:** Obesity and Metabolism Laboratory, The Institute for Drug Research, School of Pharmacy, Faculty of Medicine, The Hebrew University of Jerusalem, Jerusalem 9112001, Israel; eladbenc@gmail.com (E.B.-C.); anna.permyakova@mail.huji.ac.il (A.P.); shaharazar10@gmail.com (S.A.); shira.hircsh@mail.huji.ac.il (S.H.); saja.baraghithy@mail.huji.ac.il (S.B.); liad77@gmail.com (L.H.)

**Keywords:** CBDA, obesity, *Magel2*, PWS, hyperphagia, hepatic steatosis, dyslipidemia

## Abstract

Obesity is a global medical problem; its common form is known as diet-induced obesity (DIO); however, there are several rare genetic disorders, such as Prader–Willi syndrome (PWS), that are also associated with obesity (genetic-induced obesity, GIO). The currently available therapeutics for treating DIO and GIO are very limited, and they result in only a partial improvement. Cannabidiolic acid (CBDA), a constituent of Cannabis sativa, gradually decarboxylates to cannabidiol (CBD). Whereas the anti-obesity properties of CBD have been reasonably identified, our knowledge of the pharmacology of CBDA is more limited due to its instability. To stabilize CBDA, a new derivative, CBDA-O-methyl ester (HU-580, EPM301), was synthesized. The therapeutic potential of EPM301 in appetite reduction, weight loss, and metabolic improvements in DIO and GIO was tested in vivo. EPM301 (40 mg/kg/d, i.p.) successfully resulted in weight loss, increased ambulation, as well as improved glycemic and lipid profiles in DIO mice. Additionally, EPM301 ameliorated DIO-induced hepatic dysfunction and steatosis. Importantly, EPM301 (20 and 40 mg/kg/d, i.p.) effectively reduced body weight and hyperphagia in a high-fat diet-fed *Magel2*^null^ mouse model for PWS. In addition, when given to standard-diet-fed *Magel2*^null^ mice as a preventive treatment, EPM301 completely inhibited weight gain and adiposity. Lastly, EPM301 increased the oxidation of different nutrients in each strain. All together, EPM301 ameliorated obesity and its metabolic abnormalities in both DIO and GIO. These results support the idea to further promote this synthetic CBDA derivative toward clinical evaluation in humans.

## 1. Introduction

Obesity is a global medical problem, and treating it is a constant challenge for both the patient and healthcare system [1]. Importantly, it is observed at a higher scale with the western population, where excessive eating and a sedentary lifestyle are quite common [2,3]. This form of obesity is also known as diet-induced obesity (DIO) [4]. The first choice in treating DIO is lifestyle change in terms of nutrition and activity, although low compliance among patients is a common issue. Additional treatment options include bariatric surgery and medications; however, the success of these treatments is quite limited [5]. Other than DIO, there are also different genetic disorders related to obesity (genetic-induced obesity, GIO) [6]. One example is Prader–Willi syndrome (PWS), a multisystemic genetic disorder caused by a lack of multiple genes on the paternally inherited chromosome 15q11.2-q13 region [7]. It is characterized by a wide variety of clinical symptoms such as delayed motor and language skills, cognitive disability, hypotonia, hypogonadism, and short stature. Interestingly, the two most distinctive symptoms of PWS are excessive eating (hyperphagia), which can be followed by morbid obesity [8]; these symptoms are considered as the main treatment goals in PWS patients. The current PWS treatments are very limited: they include strict dietary supervision and growth hormone (GH) therapy, resulting in only partial improvement of these symptoms [9,10].

One of the potential therapeutic targets of DIO and GIO is the endocannabinoid system (ECS), which was found to be over-activated in patients with obesity [11], as well as in PWS patients and animal models [12]. In fact, modulation of the ECS is known for its potential to treat obesity. One promising approach is based on peripherally restricted antagonism of cannabinoid-1 receptor (CB_1_R) [13], which was found to be efficacious in ameliorating both DIO and GIO in mice [12,14]. Another possible approach involves phytocannabinoids, which are cannabinoids that originated from the *Cannabis sativa* plant. The main ones include Δ^9^-tetrahydrocannabinol (THC), which serves as a CB_1_R and CB_2_R partial agonist [15], as well as cannabidiol (CBD; Figure 1A), which serves as a CB_1_R negative allosteric modulator [16]. Its additional mechanisms include but are not limited to modulation of 5-HT1A [17] and GPR55 [18,19]. Each of these two molecules has been reported to affect various mechanisms involving obesity pathophysiology. THC treatment resulted in a reduction in food intake in animals [20], and it restored insulin sensitivity in adipocytes [21]. Despite these beneficial effects, the psychoactive effect of THC creates a problematic safety profile, which disfavors it as an effective therapy.

Conversely, CBD is non-psychoactive and, therefore, is safer to use. Different studies demonstrated the positive effects of CBD on various obesity-related mechanisms. For instance, CBD was found to reduce body weight [22] and to prevent hyperphagia [23] in rat models. Moreover, a clinical study investigating CBD administration for epilepsy reported loss of appetite as a common side effect [24]. In addition, an in vitro study showed that CBD promotes the browning of white adipocytes [25]. All together, these findings suggest that CBD has the potential to regulate whole-body energy homeostasis.

CBD is formed from a cannabidiolic acid (CBDA; Figure 1B) via gradual decarboxylation [26]. Whereas the biological and therapeutic properties of CBD in modulating obesity parameters have been reasonably identified, our knowledge of the pharmacology of CBDA is much more limited due to its instability, which further weakens the case for developing it as a medicine. In an attempt to find a more stable and effective CBDA derivative, CBDA-O-methyl ester (EPM301 (previously named HU-580; Figure 1C) was suggested as an alternative substance. This derivative was originally synthesized by the Mechoulam group in order to study the biogenesis of acidic phytocannabinoids [27]. Interestingly, EPM301 has shown impressive results in treating neuro-behavioral conditions such as anxiety and nausea, demonstrating efficacy at lower doses compared to CBDA [28], and major depression [29]. Accordingly, depression was linked to obesity in different studies [30,31,32,33]. Hence, an intriguing question is whether EPM301 can be used to treat DIO and GIO.

To address this question, we used DIO and GIO animal models, which included high-fat diet (HFD)-fed C57Bl/6J mice, and mice with a knockout of the *Magel2* gene (*Magel2*^null^), respectively. *Magel2*^null^ mice recapitulate various features of PWS [34,35], including hyperphagia, under HFD [12]. In addition, we also used liver and kidney cell cultures as a model for fat accumulation. Our results show that EPM301 ameliorates obesity and its metabolic abnormalities in mice and cells. Our findings suggest that EPM301 may have therapeutic potential for the treatment of both DIO and GIO.

## 2. Results

### 2.1. EPM301 Ameliorates Obesity and Its Metabolic Abnormalities in DIO C57Bl/6J Mice

We first tested the efficacy of EPM301 in DIO. Six-week-old C57Bl/6J male mice were fed with an HFD for 14 weeks, and then treated daily with EPM301 (40 mg/kg/day, i.p.) for 28 days. Interestingly, EPM301 reduced body weight (Figure 2A) and fat mass (Figure 2B), without any significant effect on lean mass (Figure 2C). In addition, EPM301 significantly reduced hyperleptinemia (Figure 2D), a known feature of DIO. Whole-body metabolic homeostasis was also changed in EPM301-treated mice, as evidenced by decreased fat oxidation (FO; Figure 2E) and increased carbohydrate oxidation (CHO; Figure 2F), with no effect on total energy expenditure (TEE; Figure 2G) or on ambulatory activity (Figure 2H), which was decreased following HFD feeding. However, EPM301 completely restored the voluntary wheel running ability of the mice (Figure 2I), suggesting that EPM301 may reduce obesity-induced depression symptoms.

EPM301 also improved the glycemic parameters. Although EPM301 did not ameliorate HFD-induced hyperglycemia following 24 h of fasting (Figure 3A), it improved glucose tolerance (Figure 3B,C) and significantly decreased HFD-induced hyperinsulinemia (Figure 3D), even though it had no effect on exogenous insulin tolerance (Figure 3E,F). Interestingly though, EPM301 was able to normalize HFD-induced hyperglycemia following 6 h of fasting to the levels measured in STD-fed mice (Figure 3F), suggesting a time-dependent effect of EPM301 on glucose utilization or compensatory endogenous mechanisms that may alleviate its hypoglycemic effect over time.

We next evaluated EPM301’s ability to reverse DIO-induced hepatic steatosis and liver injury. Indeed, EPM301 normalized obesity-associated elevation in liver enzyme levels (AST, ALT, and ALP; Figure 4A–C) as well as reduced liver fat accumulation and hepatic triglycerides (Figure 4D,E), demonstrating the ability of EPM301 to ameliorate DIO-induced hepatic steatosis. Similarly, EPM301 was effective in reducing circulating dyslipidemia, such as triglycerides (Figure 4F) and total cholesterol (Figure 4G). More specifically, EPM301 increased the HDL/LDL ratio (Figure 4H) with a negligible effect on the HDL levels (Figure 4I), and it significantly reduced the LDL levels (Figure 4J).

These positive in vivo effects on the liver and lipid homeostasis led us to investigate the hepatic effect in vitro on HepG2 cells. As expected, EPM301 effectively reduced lipid accumulation in the cells (Figure 5A), mostly in concentrations of 10 and 0.001 μM, suggesting a bi-phasic effect, which is common to cannabinoids [36]. We continued with these two effective concentrations of EPM301 to test the expression levels of two LDL-regulating proteins in HepG2 cells. LDL receptor (LDLR), responsible for LDL uptake [37], was upregulated by EPM301 (Figure 5B), and the PCSK9 enzyme, which causes LDLR degradation [38], was downregulated (Figure 5C) with both concentrations. Collectively, these results imply that EPM301 is hepatoprotective and can restore liver function and dyslipidemia in DIO.

Moreover, EPM301 exhibited beneficial renal effects. The reductions in the urine excretion/water consumption ratio (Figure A1A) and creatinine clearance (Figure A1B) in obese mice were reversed by EPM301 treatment, indicating preserved kidney function. In addition, EPM301 reduced the lipid accumulation in cultured human kidney cells (Figure A1C), suggesting that EPM301 may preserve kidney function possibly by reducing renal lipotoxicity [39].

All together, these results show that EPM301 is effective in ameliorating DIO and its metabolic abnormalities.

### 2.2. EPM301 Ameliorates Obesity and Hyperphagia in GIO Magel2^null^ Mice

In order to establish the optimal therapeutic dose of EPM301 in the GIO model, we measured the effect of a single dose on post-fasting cumulative food intake in C57Bl/6J mice. Doses of 20 and 40 mg/kg EPM301, i.p. elicited the highest effect of EPM301 on reducing food intake (Figure A2A,B), and were, therefore, selected for the GIO model.

*Magel*2 nullification in male mice was used for the GIO model, and their wild-type littermates were used as controls. Similarly to the DIO model, *Magel2*^null^ mice were fed ad libitum with HFD or STD for 14–16 weeks. Next, mice were started with a daily treatment with EPM301 (40 and 20 mg/kg/day, i.p.) for 28 days. Both genotypes exhibited a significant weight loss in response to EPM301 when given a dose of 40 mg/kg. However, only the *Magel2*^null^ mice showed a similar effect when given a lower dose of 20 mg/kg (Figure 6A). This higher potency in *Magel2*^null^ mice might be associated with the ability of EPM301 to prevent hyperphagia only in this genotype (Figure 6B), an effect that was not observed in their WT littermate controls. Another difference between the genotypes was observed in an indirect calorimetric test, showing that EPM301 differently affected the metabolic status of each strain. Although EPM301 induced the *Magel2*^null^ mice to channel their use of energy to FO (Figure 6C), it promoted CHO in WT mice (Figure 6D). These effects were opposite between the genotypes, so an increased parameter in one genotype was decreased in the other.

Activity profiling was used to estimate how activity contributed to the observed weight loss. Interestingly, EPM301 increased the wheel running ability of WT mice; however, this effect was not observed in *Magel2*^null^ mice, which exhibited a significantly lower ability in all groups (Figure 6E). Importantly, EPM301 slightly restored the reduced ambulatory activity in both WT and *Magel*2^null^ mice (Figure 6F).

Next, we assessed EPM301’s potential as a preventative treatment for GIO without involving HFD. To this end, we tested the long-term preventative effect of EPM301 (20 mg/kg/day, i.p.) for 18 weeks on weight and body composition in STD-fed *Magel2*^null^ mice. Untreated *Magel2*^null^ mice exhibited the highest end-point body weight. However, EPM301-treated *Magel2*^null^ mice showed both the lowest final weight and weight gain throughout the treatment (Figure 7A,B). Similarly, the highest fat mass was found in untreated *Magel2*^null^ mice; however, EPM301 prevented this increase and normalized the levels of fat mass to those of littermate WT mice (Figure 7C,D). In *Magel2*^null^ mice, the lean body mass was barely changed by EPM301 treatment and was found to be generally lower than in littermate WT mice, possibly due to their naturally smaller size (Figure 7E). However, the lean body mass percentage showed a significant increase in the treated *Magel2*^null^ mice, similar to that of the WT mice (Figure 7F). All together, these results indicate the therapeutic potential of EPM301 in GIO.

## 3. Discussion

Treating obesity has remained a high priority goal in both DIO and GIO. In GIO (e.g., PWS), the morbidity and mortality from obesity is even higher due to a significantly lower compliance with lifestyle changes, mostly as a result of uncontrollable hyperphagia. Even when a strict lifestyle is maintained in PWS patients, the increased hunger drive persists and damages the quality of life in these patients. The ECS plays a major role in regulating food intake and body weight. In obese humans and animals, an upregulation of its activity was observed, especially in terms of the upregulation of CB_1_R and/or a higher ‘tone’ of endogenous cannabinoids (eCBs) [12]. The main objective of this study was to further investigate the potential of treating obesity by modulating the ECS in a way other than using a pure CB_1_R antagonism. In this study, the synthetic CBDA-derivative EPM301 was used as such an agent.

In a DIO model, EPM301 displayed an impressive ability to prevent or mitigate various conditions associated with the metabolic syndrome. In addition to a reduction in body weight and adiposity, EPM301-treated mice displayed a reduction in hyperleptinemia, which may indicate an increase in leptin sensitivity. EPM301 also preserved glucose homeostasis, lowered dyslipidemia, as well as preserved kidney and liver function possibly via a reduction in lipid levels. These effects, seen in kidney and liver cells, imply that EPM301 directly activated these cells. Moreover, lipid reduction by EPM301 may be linked to its ability to upregulate the LDLR levels.

The GIO model, *Magel2*^null^ mice, reacted differently to EPM301, compared with their littermate WT controls or the DIO C57Bl/6J mice. Whereas all three mouse models displayed weight reduction following EPM301 treatment, the weight loss mechanisms in the *Magel2*^null^ mice were unique. In both DIO C57Bl/6J and GIO WT controls, EPM301 increased wheel running activity, whereas *Magel2*^null^ mice presented extremely low activity, typical of this strain, due to their lower muscle tone [40], similar to that of PWS patients [8]. In addition, EPM301 affected whole-body metabolic homeostasis differently in each mouse strain; EPM301-treated *Magel2*^null^ mice utilized fat oxidation as their energy source, whereas EPM301-treated WT controls and DIO C57Bl/6J utilized carbohydrate oxidation. More importantly, the HFD-fed *Magel2*^null^ mice were the only strain that displayed hyperphagia, and EPM301 was found to normalize it. All together, it seems that the EPM301 weight loss mechanism in DIO C57Bl/6J mice and the GIO WT controls is characterized by increased activity. Conversely, *Magel2*^null^ mice weight loss is characterized by a reduction in HFD-induced hyperphagia and increased fat oxidation. The latter might explain how weight gain is prevented in STD-fed *Magel2*^null^ mice, which do not normally display hyperphagia. Both doses of EPM301 had similar effects on the *Magel2*^null^ mice, whereas in their WT controls, only the higher dose of 40 mg/kg was found to be effective.

The molecular mechanism by which EPM301 induces its beneficial effects is not fully known; however, its ability to reduce anxiety and nausea has been reported to be mediated through the serotonergic 5-HT1A receptor. EPM301 was shown to enhance 5-HT1A receptor activation following agonist binding, suggesting positive allosteric activity [28]. This receptor is involved in regulating food intake [41]. However, the 5-HT1A agonism effect on hyperphagia is controversial [41,42,43]. 5-HT1A receptors were described only in the brain [44]; however, the peripheral effects of EPM301 on liver and kidney cells may suggest the involvement of additional mechanisms. As a cannabinoid molecule and a derivative of CBDA, EPM301 may share mutual mechanisms with CBD, which could explain the direct effect of EPM301 on peripheral cells and tissues. CBD was reported to reduce hyperphagia induced by 5-HT1A or CB_1_R agonists in animals [23]. Other similarities of EPM301 effects on DIO such as restoring liver function, lipid and glucose homeostasis, and hyperleptinemia are also common with CB_1_R negative allosteric modulators (e.g., CBD) or CB_1_R global or peripherally restricted antagonists (rimonabant or JD5037, respectively) [12,45,46,47]. These comparisons may explain the rationale to examine EPM301’s activity on CB_1_R as well. Although EPM301 is not a peripherally restricted compound, psychiatric side effects, which were observed with the global CB_1_R antagonist rimonabant, are not expected with EPM301 due to its reported beneficial effects in reducing anxiety and depression [28,29], as well as its probable inability to block the CB_1_R (although this was not tested here).

In conclusion, EPM301 displayed an impressive ability to ameliorate obesity and its metabolic abnormalities in DIO and GIO animal models and cells. The weight loss mechanisms under EPM301 treatment might be different and even opposing between the two mouse genotypes, demonstrated by its ability to differently affect fat and carbohydrate oxidation as well as ambulation. EPM301 not only modulated energy utilization, but also restored glucose and insulin homeostasis as well as reduced obesity-induced hepatic steatosis and liver injury, and normalized hyperlipidemia, an effect that is most likely modulated via affecting cellular hepatic LDL metabolism. Moreover, EPM301 prevented weight and fat gain in a mouse model for PWS, two main features of this syndrome. These results provide the rationale to further support the synthetic CBDA derivative, EPM301, for clinical evaluation in humans with DIO and/or GIO, especially in PWS patients, who currently have no effective therapy available. 

## 4. Materials and Methods

### 4.1. Animals and Experimental Protocol

The experimental protocol used was approved by the Institutional Animal Care and Use Committee of the Hebrew University, which is an AAALAC International accredited institution. C57Bl/6J mice (Jackson Laboratory #000664, Bar Harbor, ME, USA) or *Magel2*^null^ mice (C57Bl/6-*Magel2*^tm1Stw^/J, Jackson Laboratory #009062) and their littermate WT controls were used for the in vivo experiments. The *Magel2*^null^ mice were maintained on a C57Bl/6J background for at least 15 generations and were genotyped as previously described [35,48]. Mice carrying a paternally inherited lacZ-knocking allele were functionally null for *Magel2* and were referred to as *Magel2*^null^ (or *Magel2*^-/-^); littermates that were WT for *Magel2* were used as controls. All mice were 6-week-old males at the beginning of each experiment.

Both genotypes were fed ad libitum with either a standard diet (STD; 14% Kcal fat, 24% Kcal protein, 62% Kcal carbohydrates; NIH-31 rodent diet) or a high-fat diet (HFD; 60% Kcal fat, 20% Kcal protein, and 20% Kcal carbohydrates; Research Diet, D12492) for 14–16 weeks, during which their body weight was monitored weekly. Then, obese mice were randomly divided into the experimental groups. Treatment with EPM301 (20 or 40 mg/kg, i.p.) or vehicle (1% Tween80, 4% DMSO, 95% Saline) was conducted for 28 days, and body weight was monitored daily. Twenty-four-hour urine output and water consumption were measured (only in C57Bl/6J mice) using the CCS2000 Chiller System (Hatteras Instruments, Cary, NC, USA). Body composition was determined using the EchoMRI-100H™ (Echo Medical Systems, Houston, TX, USA). At 24 h following the last dose, the mice were euthanized by cervical dislocation under anesthesia; the blood and liver were harvested for further analyses.

For the dose calibration experiment, 12-week-old STD-fed male C57Bl/6J mice underwent a 24 h fast. At the 22nd h, mice were treated with EPM301 (0.004, 20 or 40 mg/kg, i.p.) or its vehicle. Food intake was measured constantly for 24 h from the fast end time using the Promethion High-Definition Behavioral Phenotyping System (Sable Instruments, Inc., Las Vegas, NV, USA) (further described below).

For the preventive treatment experiment, 6-week-old male STD-fed *Magel2*^null^ mice were treated with EPM301 (20 mg/kg/day, i.p.) or its vehicle for 18 weeks. Vehicle-treated WT littermates were used as controls. Mice were measured for body weight and body composition as previously mentioned.

### 4.2. Multi-Parameter Metabolic Assessment

The metabolic profiles of the mice were assessed by using the Promethion High-Definition Behavioral Phenotyping System (Sable Instruments, Inc., Las Vegas, NV, USA). Data acquisition and instrument control were performed using MetaScreen software version 2.2.18.0 (Sable Instruments, Inc., Las Vegas, NV, USA), and the obtained raw data were processed using ExpeData version 1.8.4 using an analysis script detailing all aspects of data transformation. Mice with free access to food (except in the dose selection experiment) and water were subjected to a standard 12 h light/12 h dark cycle, which consisted of a 48 h acclimation period followed by 24 h of sampling. Respiratory gases were measured by using the GA-3 gas analyzer (Sable Systems, Inc., Las Vegas, NV, USA) using a pull-mode, negative-pressure system. Air flow was measured and controlled by FR-8 (Sable Systems, Inc., Las Vegas, NV, USA), with a set flow rate of 2000 mL/min. Water vapor was continuously measured and its dilution effect on O_2_ and CO_2_ was mathematically compensated. Effective mass was calculated by ANCOVA analysis, as described previously [49], using the calculations described (Figure A3). Respiratory quotient (RQ) was calculated as the ratio between CO_2_ produced (VCO_2_) to O_2_ consumed (VO_2_) based on Equation (1):RQ = VCO_2_/VO_2_(1)

Total energy expenditure (TEE) was calculated using VO_2_ and RQ based on Equation (2):TEE = VO_2_ × (3.815 + 1.232 × RQ)(2)

The values were normalized to effective body mass, and expressed as kcal/h/kg^eff.mass^. Fat oxidation (FO) and carbohydrate oxidation (CHO) were calculated using VO_2_ and VCO_2_ based on Equations (3) and (4), respectively:FO = 1.69 × VO_2_ − 1.69 × VCO_2_(3)
CHO = 4.57 × VCO_2_ − 3.23 × VO_2_(4)

The values were normalized to effective body mass, and expressed as g/d/kg^eff.mass^.

### 4.3. Locomotor Activity

Locomotor activity was quantified by the number of disruptions of infrared XYZ beam arrays with a beam spacing of 0.25 cm in the Promethion High-Definition Behavioral Phenotyping System (Sable Instruments, Inc., Las Vegas, NV, USA).

### 4.4. Glucose Tolerance Test (ipGTT) and the Insulin Tolerance Test (ipITT)

Mice that were fasted overnight were injected with glucose (1.5 g/kg i.p.), followed by a tail blood collection at 0, 15, 30, 45, 60, 90, and 120 min. Blood glucose levels were determined using the Contour^®^ glucometer (Bayer, Pittsburgh, PA, USA). On the following day, the mice were fasted for 6 h before receiving insulin (0.75 U/kg, i.p.; Actrapid vials, Novo Nordisk A/S, Bagsværd, Denmark), and blood glucose levels were determined at the same intervals as described above.

### 4.5. Blood and Urine Biochemistry

Serum and urine levels of creatinine as well as the serum levels of cholesterol, triglycerides (TG), high-density lipoprotein (HDL), alanine aminotransferase (ALT), aspartate aminotransferase (AST), alkaline phosphatase (ALP), and glucose were determined using the Cobas C-111 chemistry analyzer (Roche, Switzerland). Low-density lipoprotein (LDL) levels were calculated by using the following equation: LDL-C = (0.9 × TCHOL) − (0.9 × TG/5) − 28. Creatinine clearance was calculated using the urine and serum creatinine levels (CCr mL/h = Urine creatinine mg/dL × Urine volume/Serum creatinine mg/dL × 24 h). Fasting blood glucose was measured using the Contour^®^ glucometer (Bayer, Pittsburgh, PA, USA). Serum insulin was determined using an Ultra-Sensitive Mouse Insulin ELISA kit (Crystal Chem, Inc., Elk Grove Village, IL, USA or Millipore, Darmstadt, Germany). Serum leptin was determined by ELISA (Millipore, Darmstadt, Germany).

### 4.6. Hepatic Triglyceride Content

Liver tissue was extracted as previously described [14], and its TG content was determined using an EnzyChrom^TM^ Triglyceride assay kit (BioAssay Systems, Hayward, CA, USA).

### 4.7. Histopathology

First, 5 µm paraffin-embedded liver sections from 5 animals per group were stained with hematoxylin-eosin staining. Liver images were captured with a Zeiss AxioCam ICc5 color camera (Carl Zeiss AG, Jena, Germany) mounted on a Zeiss Axio Scope.A1 light microscope (Carl Zeiss AG, Jena, Germany) and taken from 10 random 40× fields of each animal.

### 4.8. Cell Culture

Liver HepG2 (HB-8065, ATCC) and kidney HK-2 (CRL-2190, ATCC) cells were cultured in RPMI-1640 medium (Biological Industries, Beit HaEmek, Israel) containing 10% fetal bovine serum (FBS; Cat# 12657, Gibco Biosciences, Dublin, Ireland) or in a low-glucose DMEM medium (Biological Industries, Beit HaEmek, Israel) containing 5% FBS. Both media also contained 2 mM l-glutamine, 1 mM sodium pyruvate, 100 U/mL penicillin, and 100 mg/mL streptomycin. Cells were incubated at 37 °C in a humidified atmosphere of 5% CO_2_/95% air. Cell experiments were conducted at >80% confluence in 96-well plates, 6-well plates, or 10 cm plates as follows. 

#### 4.8.1. CBDA-Derivative Treatment

Cells were cultured in a serum-free RPMI-1640 medium containing the CBDA-derivative EPM301 (or its vehicle, DMSO as a negative control) in a range of concentrations (as detailed in the results section).

#### 4.8.2. Lipotoxic Conditions

HepG2 and HK-2 cells were cultured in FBS-free versions of RPMI-1640 medium or in a low-glucose DMEM medium. Both media also contained a mixed solution of sodium oleate (Cat# O7501; Sigma–Aldrich, St. Louis, MO, USA) and sodium palmitate (Cat# P9767; Sigma–Aldrich) at a ratio of 2:1, at a final concentration of 0.3 mM.

#### 4.8.3. Cellular Fat Accumulation

To determine whether EPM301 has the ability to reduce lipid accumulation in liver or kidney cells, we utilized an in vitro model of fat accumulation in HepG2 or HK-2 cells. Briefly, HepG2 or HK-2 cells were exposed to lipotoxic conditions as mentioned above. Control cell cultures were incubated with media containing the vehicles. EPM301 was tested in a wide range of concentrations (as detailed in the results section) in the presence/absence of fatty acids. After 24 h of incubation with the compounds, the cells were washed, incubated with a 1 μg/mL mixture of Nile Red/Hoechst solution for 15 min at 37 °C and protected from light. Fluorescence was measured by the SpectraMax iD3 microplate reader (Molecular Devices, LLC., San Jose, CA, USA) at ex: 488 nm/em: 550 nm for Nile Red and Hoechst, respectively. Results were normalized to the total protein levels of each well and presented as a change in the accumulation of lipids in comparison with the vehicle-treated group.

### 4.9. Western Blotting

HepG2 were harvested by trypsin EDTA or scraping, precipitated in 1× PBS, and then resuspended in a RIPA buffer. Protein concentrations were measured with a Pierce BCA Protein assay kit (Thermo Fisher Scientific, Waltham, MA, USA). Samples were resolved by SDS–PAGE (4–15% acrylamide, 150 V) and transferred to PVDF membranes using a Trans-Blot Turbo Transfer System (Bio-Rad, Hercules, CA, USA). Membranes were then incubated for 1 h in 5% milk (in 1× TBS-T) to block unspecific binding. Blots were incubated overnight with primary antibodies against LDL receptor (LDLR) (1:1000; Cat# ab52818, Abcam, Cambridge, UK), proprotein convertase subtilisin/kexin type-9 (PCSK9)_(human)_ (1:1000; Cat# ab181142, Abcam) at 4 °C, or with β-actin horseradish peroxidase (HRP)-conjugated primary antibody (1:30000; Cat# ab49900, Abcam) for 1 h at room temperature. Anti-rabbit (1:2500; Cat# ab97085, Abcam) HRP-conjugated secondary antibody was used for 1 h at room temperature, followed by chemiluminescence detection using a Clarity Western ECL Blotting Substrate (Bio-Rad). Densitometry was quantified using ImageJ and Image Lab software (NIH, USA).

### 4.10. Statistics

Values are expressed as the mean ± SEM. An unpaired two-tailed Student’s *t*-test was used to determine the differences between the two groups. Results in multiple groups and time-dependent variables were compared by ANOVA, followed by a Tukey’s multiple comparisons test. Significance was set at *p* < 0.05.

## 5. Patents

The use of EPM301 for treating DIO and GIO is protected by a few patents: 62/522,243; 62/903,852; and 63/221,006.

## Figures and Tables

**Figure 1 ijms-23-05610-f001:**
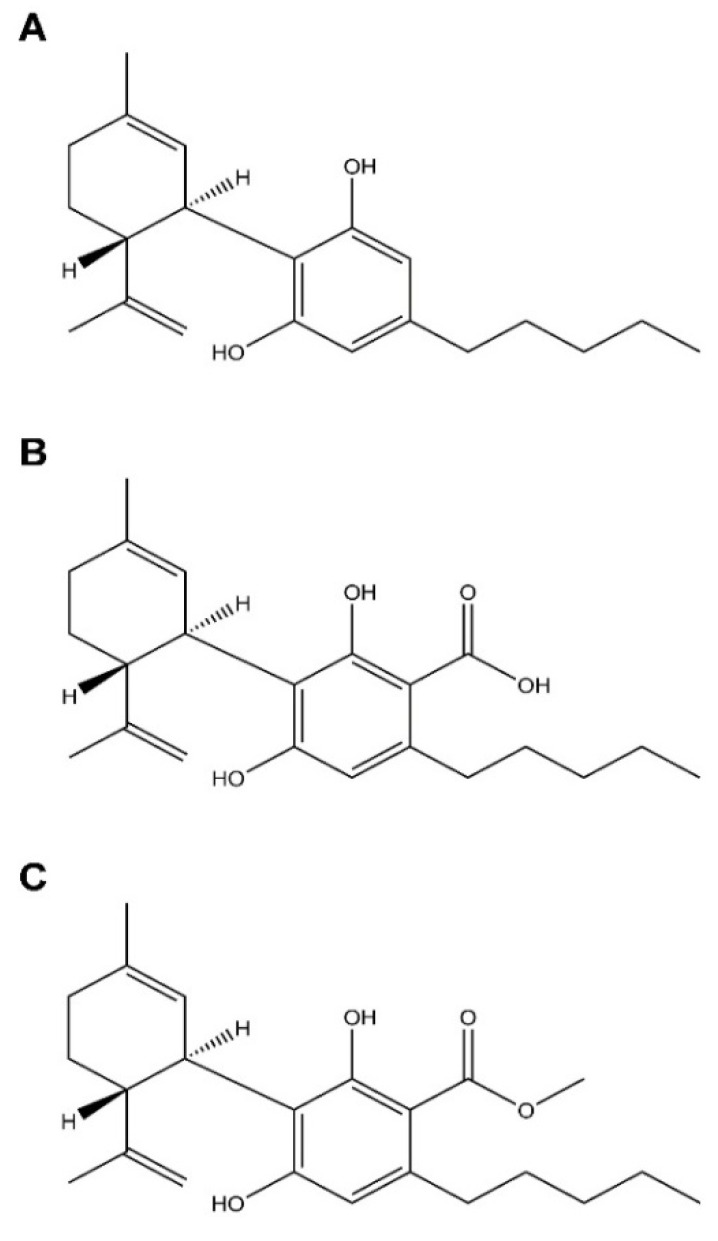
The molecular structures of CBD (**A**), CBDA (**B**), and CBDA-O-methyl ester (EPM301) (**C**).

**Figure 2 ijms-23-05610-f002:**
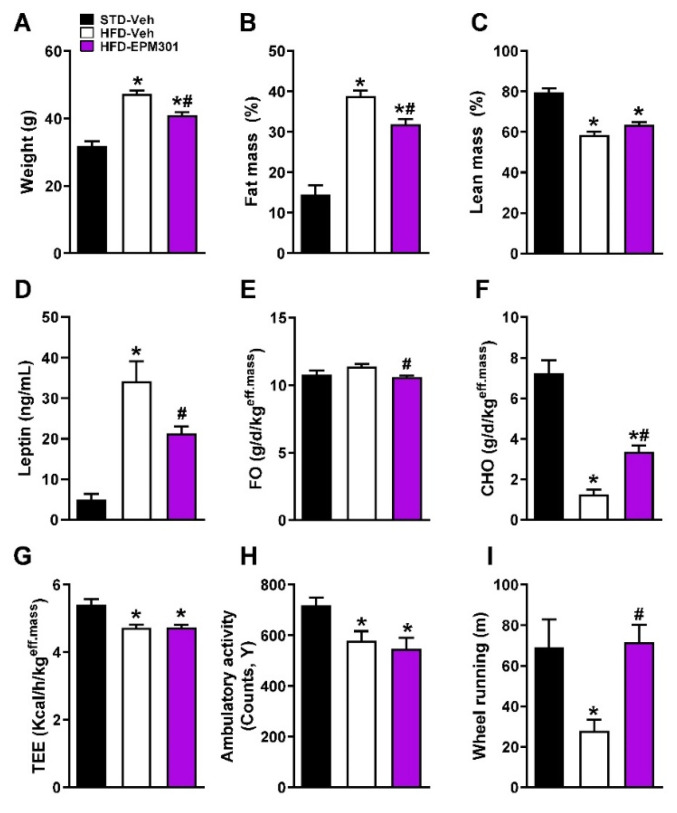
EPM301 reduces body weight, adiposity, and hyperleptinemia and affects energy and activity parameters in DIO C57Bl/6J mice. EPM301 (40 mg/kg/day, i.p.) for 28 days reduced body weight (**A**) and fat mass (**B**) with a slightly insignificant increase in lean mass (**C**). These effects were associated with reducing serum leptin levels (**D**). The treatment also reduced fat oxidation (**E**) and increased carbohydrate oxidation (**F**), without affecting the total energy expenditure (**G**). In addition, although EPM301 did not significantly affect the ambulatory activity (**H**), it still normalized voluntary wheel running (**I**). Data represent the mean ± SEM from 5–6 mice per group. * *p* < 0.05 relative to STD-Veh; # *p* < 0.05 relative to HFD-Veh.

**Figure 3 ijms-23-05610-f003:**
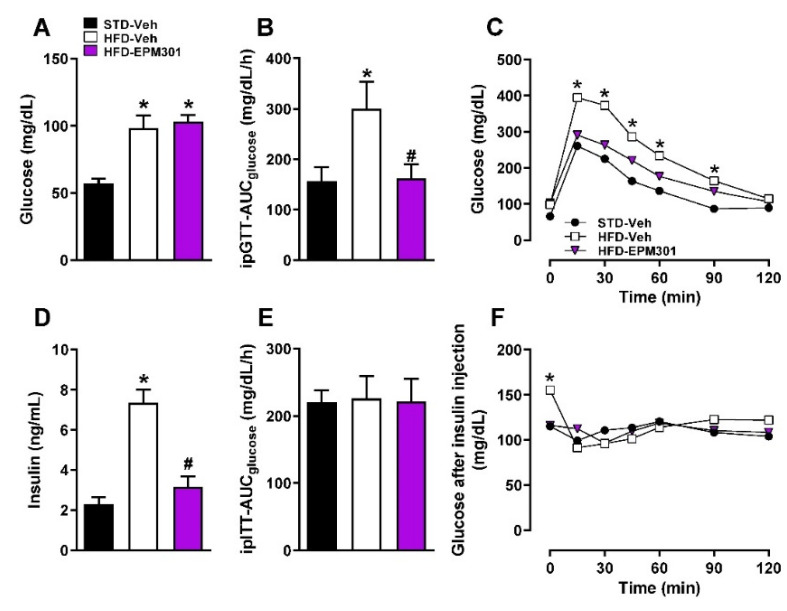
EPM301 improves glucose homeostasis in DIO C57Bl/6J mice. Daily chronic treatment of HFD-induced obese C57Bl/6J mice with EPM301 (40 mg/kg/day, i.p.) for 28 days did not attenuate the HFD-induced hyperglycemia (**A**); however, it improved glucose tolerance (**B**,**C**) and attenuated hyperinsulinemia (**D**). Insulin tolerance was not affected (**E**,**F**). Data represent the mean ± SEM from 5–6 mice per group. * *p* < 0.05 relative to STD-Veh; # *p* < 0.05 relative to HFD-Veh.

**Figure 4 ijms-23-05610-f004:**
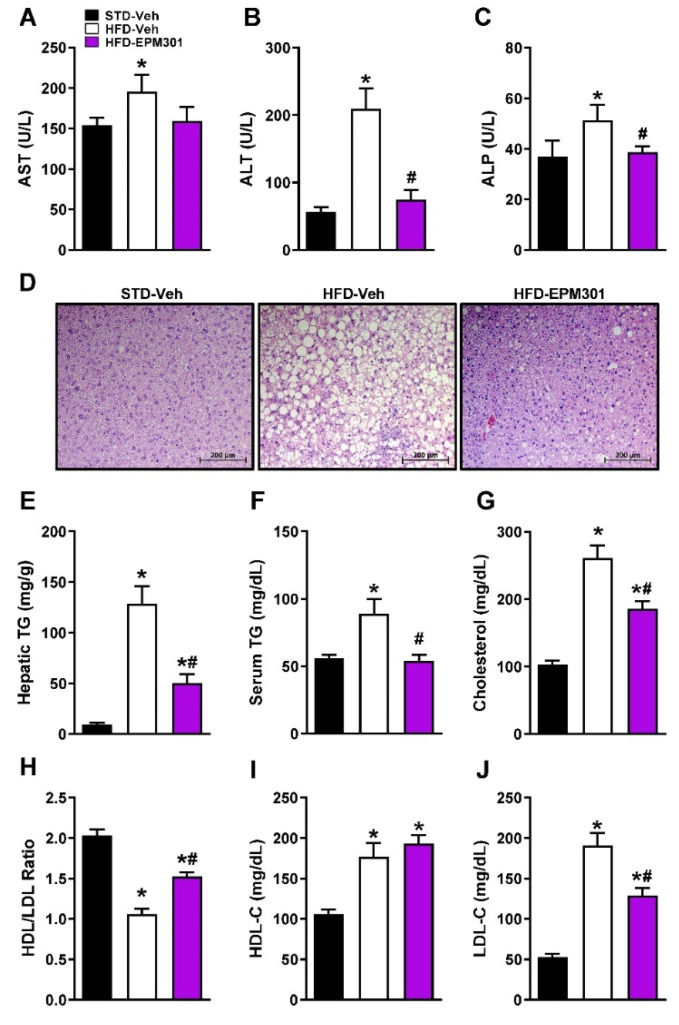
EPM301 restores liver function and lipid homeostasis in DIO C57Bl/6J mice. Daily chronic treatment of HFD-induced obese C57Bl/6J mice with EPM301 (40 mg/kg/day, i.p.) for 28 days reduced the HFD-induced hepatic injury and steatosis, as manifested by the reduced serum levels of AST (**A**), ALT (**B**), and ALP (**C**), along with a reduction in lipid vacuoles in hepatocytes (**D**) and hepatic triglyceride (TG) content (**E**). EPM301 improved the serum lipid profile, as manifested by reduced serum TG (**F**) and total cholesterol (**G**). EPM301 increased the HDL/LDL ratio (**H**) without changing the HDL levels (**I**) but still reduced the LDL levels (**J**). Data represent the mean ± SEM from 5–6 mice per group. * *p* < 0.05 relative to STD-Veh; # *p* < 0.05 relative to HFD-Veh.

**Figure 5 ijms-23-05610-f005:**
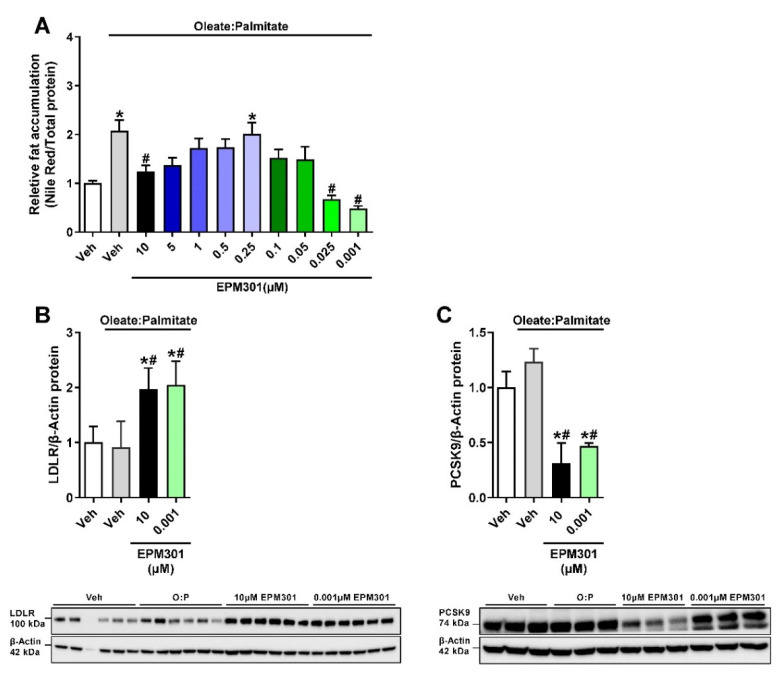
EPM301 reduces fat accumulation and affects the expression of LDL-regulating proteins in HepG2 liver cells. Treatment of fatty acid-exposed HepG2 cells with EPM301 in concentrations of 10 and 0.001 μM reduced fat accumulation, as shown by a Nile Red test (**A**). EPM301 also upregulated the LDLR protein expression levels (**B**) and downregulated the PCSK9 expression levels (**C**) in these cells, as measured by Western blotting. Data represent the mean ± SEM from 3–8 biological replicates per group. * *p* < 0.05 relative to Veh; # *p* < 0.05 relative to Veh + O:P.

**Figure 6 ijms-23-05610-f006:**
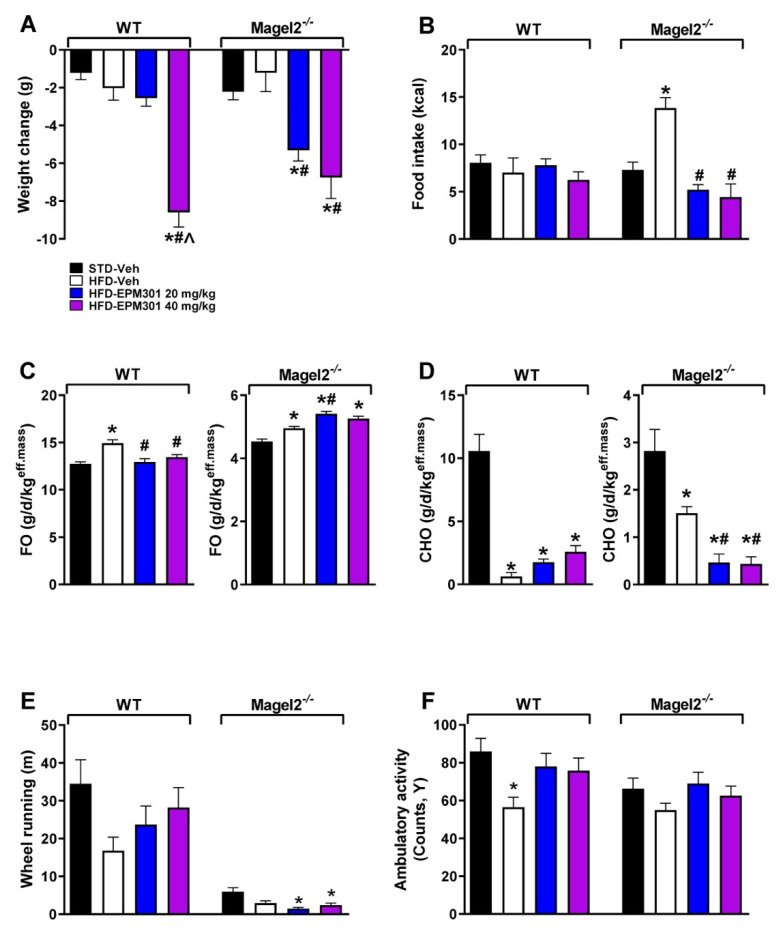
EPM301 promotes weight loss, decreases food intake, changes the energy profile, and affects activity in GIO *Magel2*^null^ mice. Daily chronic treatment of obese *Magel2*^null^ mice with EPM301 (20 or 40 mg/kg/day, i.p.) for 28 days reduced body weight at day 28, compared to day 1 (**A**), and caloric food intake (**B**). EPM301 also increased fat oxidation (FO; **C**) in these mice and reduced carbohydrate oxidation (CHO; **D**). Wheel running was not affected and was generally low among all groups in this genotype (**E**); however, the ambulatory activity slightly increased (**F**). Data represent the mean ± SEM from 4–13 mice per group. * *p* < 0.05 relative to STD-Veh of the same strain; # *p* < 0.05 relative to HFD-Veh of the same strain, ^ *p* < 0.05 relative to HFD-EPM301 20mg/kg of the same strain.

**Figure 7 ijms-23-05610-f007:**
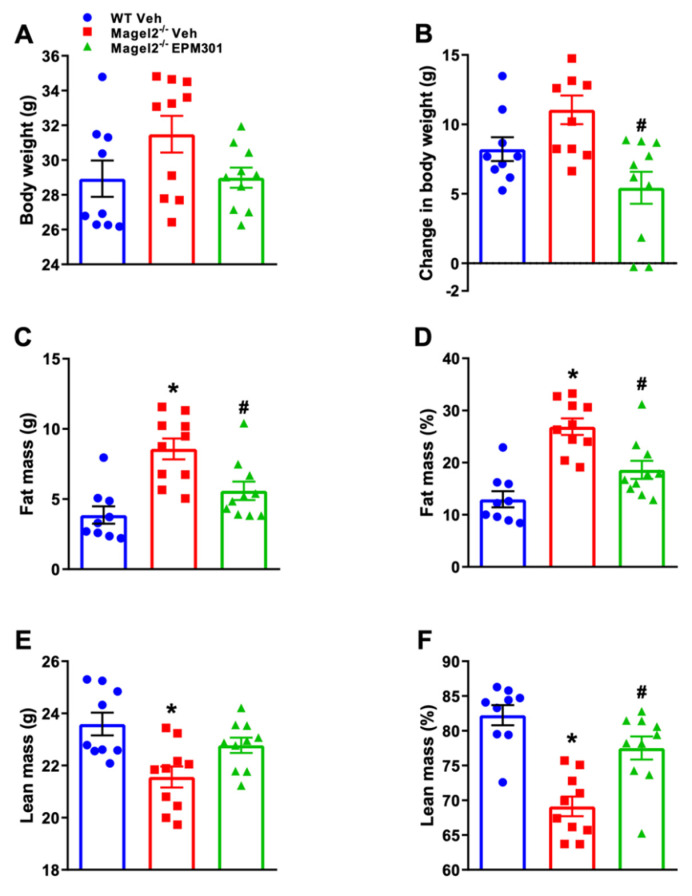
EPM301 prevents weight gain and adiposity in STD-fed GIO *Magel2*^null^ mice. Daily chronic treatment of STD-fed *Magel2*^null^ mice with EPM301 (20 mg/kg/day, i.p.) for 18 weeks prevented an increase in body weight (**A**,**B**) and fat mass (**C**,**D**). The absolute value of lean mass was not affected by EPM301 (**E**), but the lean percentage was increased back to the norm (**F**). Data represent the mean ± SEM from 9–10 mice per group. * *p* < 0.05 relative to WT-Veh; # *p* < 0.05 relative to *Magel2*^null^-Veh.

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
