# Peer review of "The Metabolic Efficacy of a Cannabidiolic Acid (CBDA) Derivative in Treating Diet- and Genetic-Induced Obesity"

_ijms, 2022, doi:10.3390/ijms23105610_

Round 1
Reviewer 1 Report
In this study proposed by Ben-Cnaan et al., the efficacy of a phytocannabinoids (Cannabidiolic acid-CBDA) as a potential therapeutic target on two forms of obesity (Diet-Induced Obesity DIO or Genetic-Induced obesity GIO) has been studied.
The first part of the studies is carried out on a DIO model, HFD-fed mice. Authors show that after a 14-16 week feeding period, animals EPM301 (a CBDA derivative) is able to reduce body weight, fat mass and to modify energy parameters. This is associated with metabolic improvements (lower plasma insulin and better glucose homeostasis) and reduction of steatosis. Authors also observed a greater plasma cholesterol profile that can be explained by both increase of LDLr and decrease of PCSK9.
In a second part of the study, a GIO model of Prader-Willi Syndrome, the Magel2-/- mouse has been used. The same way as DIO mice, EPM 301 promotes weight loss by reducing food intake and changes in energy profile when mice are fed a HFD diet.
Interestingly, in a last part, authors demonstrate that EPM301 is able to prevent weight gain and adiposity mice are treated with a chronic treatment for 18 weeks and fed a Standard diet.
The study addresses the interesting problem of potential therapeutic target in obesity. The study is well designed and provide interesting data. Results are well interpreted and the discussion is correctly constructed.
Meanwhile, some points could be addressed to strengthen the manuscript.
- One of the mechanism of EPM301 is to reduce food intake. It has been clearly shown in figure A2. Meanwhile, the DIO model, food intake has not been measured. Some conclusions can be done using leptinemia but no direct measurements. Why food intake has not been measured? Is there any evidence in literature or in personal authors data that the drug has such effects in DIO model?
- Impact of EPM301 on metabolic profile is very interesting, especially on glucose homeostastis. Meanwhile, the gold standard glucose tolerance test is not i.p. but oral GTT. Using i.p. shunt the incretins secretion and give results that can be different from physiology reality. Is there any reasons for specifically using i.p. GTT ?
- Fasted plasma glucose (overnight) is more elevated in HFD and is not influenced by EPM301 treatment (as shown in figure 3A and C). When mice are fasted for 6h, for ITT test, glycemia of HDF-EPM301 group becomes not different from control and lower than HFD group. This result should be discussed.
- In figure 4, total cholesterol concentration of STD-veh group is 100mg/dl, but HDL-c is 100mg/dl and LDL-c is 50 mg/dl. The same for all groups. How is it possible when adding LDL+HDL to obtain more than total?
- Results on kidney are interesting but I do not understand the usefulness of these data in this context and what additional arguments they provide. Moreover, the figure is in the supplemental data section.
- In many figures, legends are very difficult to read. Figure 6 for exemple the blue and pink bars have the same name. It is difficult at the first sight to know what dose is used. The same for the figure A2 with three EPM301 groups and 2doses presented the legend.
Reviewer 2 Report
The effect of the methyl ester form of cannabidiolic acid (EPM301) is tested on obesity using DIO and GIO mice models and different cell types. Authors found EPM301ameliorates obesity and related metabolic abnormalities in these animal models and cells.
The manuscript can be accepted after fixing the following minor comments
1) Authors should mention the details of mice treatment with EPM301 in the methods.
2) In Western blotting, authors should mention the cell type used in experiment.
3) Conclusions should be expanded.
